# TreeDQN: Sample-Efficient Off-Policy Reinforcement Learning for Combinatorial Optimization

## Abstract

A convenient approach to optimally solving combinatorial optimization tasks is the Branch-and-Bound method. The branching heuristic in this method can be learned to solve a large set of similar tasks. The promising results here are achieved by the recently appeared on-policy reinforcement learning (RL) method based on the tree Markov Decision Process (MDP). To overcome its main disadvantages, namely, very large training time and unstable training, we propose TreeDQN, a sample-efficient off-policy RL method trained by optimizing the geometric mean of expected return. To theoretically support the training procedure for our method, we prove the contraction property of the Bellman operator for the tree MDP. As a result, our method requires up to 10 times less training data and performs faster than known on-policy methods on synthetic tasks. Moreover, TreeDQN significantly outperforms the state-of-the-art techniques on a challenging practical task from the ML4CO competition.

## 1 Introduction

Industrial tasks in multiple areas such as logistics (Bertsimas & Van Ryzin, 1991), portfolio management (Markowitz, 1952), manufacturing (Barahona et al., 1988), etc., can be formulated as combinatorial optimization problems, such as Mixed Integer Linear Programs (MILP), (Wolsey & Nemhauser, 1999). Solving a large set of similar MILP tasks with varying parameters corresponding to different numbers of vehicles, customer demands, locations, etc., is usually necessary. If each task is solved independently, the optimal solution can be efficiently obtained with the *Branch-and-Bound* algorithm (B&B) (Land & Doig, 2010). The B&B algorithm employs *divide-and-conquer* approach. It iteratively builds a tree, where the root node corresponds to the initial problem, and child nodes correspond to problems with the restricted domains. The performance of the B&B algorithm hardly depends on two sequential decision-making processes: variable selection and node selection. Node selection picks the next node in the B&B tree to evaluate, and variable selection chooses the next variable to split the domain (Linderoth & Savelsbergh, 1999). The efficiency of the solver depends on the number of the destination/leaf nodes of the tree. The variable selection method (*branching rule*) should produce trees of the smallest possible size. Although the optimal branching rule is unknown (Lodi & Zarpellon, 2017), modern solvers implement human-crafted heuristics designed to perform well on a wide range of tasks (Gleixner et al., 2021).

Suppose a set of similar tasks with varying parameters is solved frequently. In that case, the B&B method can be significantly accelerated if a branching rule is adapted to the distribution of tasks by using the reinforcement learning (RL) paradigm (Etheve et al., 2020; Scavuzzo et al., 2022). To solve a MILP task, the B&B method generates a tree of nested subtasks. The typical RL algorithms are designed to process sequential data Sutton & Barto (2018). Hence, they cannot be directly applied to the tree structure in the B&B algorithm. Indeed, the RL method, which learns optimal branching decisions, should map this tree structure to an episode. A promising approach to such mapping is to consider the decision-making process of a tree MDP instead of a temporal MDP Scavuzzo et al. (2022). However, there exist several difficulties of known RL methods to the variable selection process Etheve et al. (2020); Scavuzzo et al. (2022), namely: 1) the computational complexity of MILP is very high, which limits the applicability of on-policy algorithms such as tMDP+DFS Scavuzzo et al. (2022); 2) it is challenging to accurately predict the resulting tree sizes, whose distributions

usually have extremely high variance; and 3) there is no theoretical evidence of convergence for tree MDP of such methods as FMSTS Etheve et al. (2020).

To overcome these difficulties, this paper presents a novel RL method. In particular, our contribution is as follows:

1. We present a novel sample efficient off-policy RL algorithm for the tree Markov Decision Process, TreeDQN, which is trained significantly faster than existing on-policy techniques for combinatorial optimization.

2. To overcome the issue with high variance of tree size distribution, we propose a loss function that optimizes the geometric mean of expected return.

3. We provide a theoretical basis for using the Bellman operator to train the RL agent in our TreeDQN by proving that the Bellman operator in the tree Markov Decision Process is contracting in mean.

4. We demonstrate our method's superior performance and learning efficiency on a set of synthetic and practical MILP tasks. In particular, our TreeDQN is the first successful RL algorithm to learn an efficient variable selection policy for the complex practical task from the ML4CO competition (Gasse et al., 2022).

## 2 BACKGROUND

We formally define the task as solving a set of similar MILP non-convex optimization problems:

$$\min_x \Big\{ c^\top x \colon Ax \le b \,, x \in \big[ l, u \big] \,, x \in \mathbb{Z}^m \times \mathbb{R}^{n-m} \Big\}. \tag{1}$$

It is assumed that objective coefficient vectors $c \in \mathbb{R}^n$, right-hand-side constraints $b \in \mathbb{R}^m$, constraint matrices $A \in \mathbb{R}^{m \times n}$, lower and upper bound vectors $l \in \mathbb{R}^n$, $u \in \mathbb{R}^n$, and integrality constraints $m \ge 1$ are taken from a corresponding joint probability distribution. A most commonly used method of finding the optimal solution for each MILP problem with fixed $c, b, A, l, u, m$ is the B&B (Alg. 2 in Appendix A) (Land & Doig, 2010). It builds a tree of nested MILP subproblems with non-overlapping feasibility sets. The algorithm computes a lower bound (LB, dual bound) for each subproblem and updates a global upper bound (GUB, primal bound) for the whole space of solutions. The lower bound is an optimal solution for LP relaxation of the subproblem. LP relaxation considers all discrete variables continuous and maintains all the other constraints. The GUB is the minimum over the found feasible solutions. A solution of LP relaxation is feasible if it satisfies the integrality constraints of the original problem. B&B uses these bounds to enforce its efficiency by pruning the tree. Pruning discards subtrees that can not contain a feasible solution better than the current GUB. Visiting every open node guarantees that the B&B eventually finds the best integer-feasible solution. The efficiency of the B&B depends on the *variable selection process (branching rule)*, which selects an integer variable for splitting, and the *node selection strategy*, which arranges the open leaves for visiting.

To solve the task (1), RL techniques have recently been implemented to optimize these strategies Etheve et al. (2020). Let us discuss the connections between the variable selection process and the Markov Decision Process (MDP). Usually, MDP is defined by the tuple $(\mathcal{S}, \mathcal{A}, p_{init}, p, r)$, where $\mathcal{S}$ is the set of states, $\mathcal{A}$ is the set of actions, $p_{init}(s_0)$ is the distribution of initial states, $p(s'|s, a)$ is transition probability i.e. probability of transitioning to state $s' \in \mathcal{S}$ if taken action $a \in \mathcal{A}$ in a state $s \in \mathcal{S}$ and $r(s, a, s')$ is the reward function. Although the next state $s'$ can be chosen stochastically, in each transition, state action pair $(s, a)$ should have exactly one next state: $\sum_{s' \in \mathcal{S}} p(s'|s, a) = 1$. The Markovian property says that $p(s'|s, a)$ should be a function of state, and action, and $r(s, a, s')$ should be a function of state, action, and next state. So, the episode in such MDP will have a linear structure containing tuples of $(s, a, s')$. Here, we will use the extended formulation of MDP called tree MDP (Scavuzzo et al., 2022). Unlike the usual formulation of (temporal) MDP, in tree MDP, each state can have multiple next states, i.e., $\sum_{s' \in \mathcal{S}} p(s'|s, a) \ge 1$, so the episodes will have a tree structure. Since the B&B algorithm splits the domain of an integer variable into two parts, we can formally define the variable selection process as a tuple $(\mathcal{S}, \mathcal{A}, p_{\text{init}}, p^+, p^-, r)$, where state $s \in \mathcal{S}$ is $(\text{MILP}_t, \text{GUB}_t)$, action $a \in \mathcal{A}$ is the fractional variable chosen for splitting, $p_{\text{init}}(s_0)$ is the initial

state distribution of MILP tasks, $p^+(s_{t+1}^+|s_t, a_t)$ and $p^-(s_{t+1}^-|s_t, a_t)$ denotes probabilities of visiting left ($s_{t+1}^+$) and right ($s_{t+1}^-$) next states and $r : S \to \mathbb{R}$ is the reward function. MILP$_t$ is defined as a bipartite graph in which the edges correspond to connections between constraints and variables. The tree MDP defines the value function $V(s)$ as a sum of reward and value functions of the next states:

$$V^\pi(s_t) = r(s_t, a_t, s_{t+1}^\pm) + p^+ V^\pi(s_{t+1}^+) + p^- V^\pi(s_{t+1}^-) \tag{2}$$

The next states $s_{t+1}^+$, $s_{t+1}^-$ are unambiguously determined by the state $s_t$ and action $a_t$. Probabilities $p^+$, $p^-$ are defined by the *node selection strategy*. Their values indicate the likelihood of visiting the corresponding state. The episode ends when the agent reaches a terminal state containing the optimal solution. The goal of the agent is to find a policy that would maximize the expected return. For instance, if the reward at each step equals $-1$, the value function equals the tree size with a negative sign. Hence, the agent maximizing the expected return would minimize the tree size. Tree MDP provides an efficient mapping between the variable selection process employed by the B&B algorithm and the Markov Decision Process, which can be used to train a reinforcement learning agent. The remaining task is constructing an efficient and stable sample training algorithm.

## 3 RELATED WORK

### 3.1 HEURISTIC METHODS

Practical implementations of the B&B algorithm in SCIP (Bestuzheva et al., 2021) and CPLEX (Cplex, 2009) solvers rely on handcrafted heuristics for node selection and variable selection. A straightforward strategy for node selection is *Depth-First-Search* (DFS), which aims to find any integer feasible solution faster to prune branches that do not contain a better solution. The default node selection heuristic in the SCIP tries to estimate the node with the lowest feasible solution. One of the best-known heuristics for the variable selection is *Strong Branching*. It is a tree-size efficient and computationally expensive branching rule (Achterberg, 2007). For each fractional variable with integrality constraint, Strong Branching computes the lower bounds for the left and right child nodes and uses them to choose the variable for splitting.

### 3.2 SUPERVISED LEARNING

A statistical approach to learning a branching rule was applied in Khalil et al. (2016) for the first time. Authors used SVM (Cortes & Vapnik, 1995) to predict the variable ranking of an expert for a single task instance. Later works (Khalil et al., 2017) and (Selsam et al., 2018) proposed methods based on Graph Convolutional Networks (GCNN) (Kipf & Welling, 2017) to find an approximate solution of combinatorial tasks. In Gasse et al. (2019), authors applied the same neural network architecture to imitate the Strong Branching heuristic in sophisticated SCIP solver (Bestuzheva et al., 2021). The imitation learning agent can not produce trees shorter than the expert. However, it solves the variable selection task much faster, especially if running on GPU, thereby significantly speeding up the whole B&B algorithm. In Gupta et al. (2020), the authors examined the choice of architectures and proposed a hybrid model that combines the expressive power of GCNN with the computational efficiency of multi-layer perceptrons. Despite the decrease in running time, imitation learning cannot lead to better heuristics.

### 3.3 REINFORCEMENT LEARNING

RL is a promising direction to learn a variable selection rule for the B&B algorithm. It keeps the guarantees of the B&B method to find an optimal solution and can significantly accelerate the algorithm by optimal choices of branching variables. A natural minimization target for an agent in the B&B algorithm is the size of the resulting tree. One of the main challenges here is to map the variable selection process to the MDP and preserve the Markov property. In the B&B search trees, probabilities of visiting descendant nodes $p^+$, $p^-$ depend on the global upper bound that can be changed by the future branching decisions, which violate the Markov property. To learn a branching rule, the FMSTS (Fitting for Minimizing the SubTree Size) algorithm was introduced in Etheve et al. (2020). In their method, an agent plays an episode until termination, fitting the Q-function to the bootstrapped return. The authors used the DFS node selection strategy to enforce the

Markov property during training. This method is sample efficient since training data can be sampled from a buffer of past experiences. However, it may not converge to the optimal policy because its training data was obtained by older and less efficient versions of the Q-function. In Scavuzzo et al. (2022), the authors proposed setting the global upper bound to the optimal solution for a MILP as an alternative method to ensure the Markov property. They derived the policy gradient theorem for the tree MDP (tMDP) and evaluated the REINFORCE-based agent on challenging tasks similar to (Gasse et al., 2019). This method could learn an optimal policy because the agent uses only the latest data. Still, it is sample inefficient since a single gradient step of the REINFORCE agent requires solving a batch of MILP tasks. Both works (Etheve et al., 2020; Scavuzzo et al., 2022) use the cumulative return to update the agent. We improve their approaches in terms of sample efficiency and agent performance. We apply the tree Bellman operator to train the agent efficiently with the buffer of previous experiences. We use a novel learning objective that stabilizes the training in the presence of high-variance returns.

# 4 OUR METHOD

In this section, we prove the "contraction in mean" property of the tree Bellman operator and introduce our sample-efficient RL method for finding the optimal solution of a MILP task.

## 4.1 CONTRACTION IN MEAN PROPERTY OF THE TREE BELLMAN OPERATOR

From the theoretical point of view, RL methods converge to an optimal policy due to the contraction property of the Bellman operator (Jaakkola et al., 1993). To apply RL to the tree MDP, we need to justify the contraction property of the tree Bellman operator (Borovkov, 2013).

**Definition 4.1.** We consider operator $T$ is *contracting in mean* if:

$$\begin{aligned} \|TV - TU\|_\infty &= p \cdot \|V - U\|_\infty, \\ \mathbb{E}\,p &< 1, \end{aligned} \tag{3}$$

where $\mathbb{E}\,p$ is the expected value of random variable $p$ and the infinity norm is defined as follows:

$$\|V - U\|_\infty = \max_{s \in \mathbb{S}} |V(s) - U(s)| \tag{4}$$

**Theorem 4.2.** *Tree Bellman operator is contracting in mean.*

*Proof.* Bellman operator for a tree MDP is defined similarly to a temporal MDP:

$$T^\pi V(s)) = r(s, \pi(s)) + \gamma \left[ p^+ V(s^+) + p^- V(s^-) \right]. \tag{5}$$

We assume the probability of having a left ($p^+$) and a right ($p^-$) child does not depend on the state. This assumption is close to the B&B tree pruning process, where the decision depends on the global upper bound. Using the definition of tree Bellman operator (Eq. 5) and the definition of the infinity norm (Eq. 4), we derive the following inequality:

$$\|T^\pi V(s) - T^\pi U(s)\|_\infty = \gamma \|p^+ V(s^+) + p^- V(s^-) - p^+ U(s^+) - p^- U(s^-)\|_\infty =$$

$$\gamma \max_{s^\pm \in \mathbb{S}} \left[ p^+ |V(s^+) - U(s^+)| + p^- |V(s^-) - U(s^-)| \right] \leq \gamma (p^+ + p^-) \max_{x \in \mathbb{S}} |V(x) - U(x)|$$

In a finite rooted tree with $K$ nodes, every node except the root has exactly one incoming edge. Hence, the number of edges is one less than the number of nodes. So the expected number of child nodes $\mathbb{E}(p^+ + p^-) = \frac{K-1}{K} < 1$. This leads to the following equations:

$$\begin{aligned} \|TV - TU\|_\infty &= (p^+ + p^-) \cdot \|V - U\|_\infty, \\ \mathbb{E}(p^+ + p^-) &< 1, \end{aligned}$$

that meets the definition of contraction in mean (Eq. 3).

$\square$

This theorem supports the assumption that our method, which utilizes the Bellman update operator, should be able to learn the optimal variable selection policy.

## 4.2 LOSS FUNCTION

RL methods generally regress the expected return with the mean squared error (MSE) loss function, thereby optimizing the prediction of the arithmetic mean. When solving a MILP problem, non-optimal branching decisions lead to weakly pruned trees with a size growing exponentially as a function of the number of integer-valued variables. As a result, the distribution of tree sizes produced by the B&B method will have a long tail (Fig. 3 in Appendix B). As the standard metric to compare the average performance of different branching rules is the geometric mean of the final tree size and execution time Gasse et al. (2019), we propose to optimize the geometric mean of the expected return, i.e., use mean squared logarithmic error (MSLE) instead of MSE. For a variable $y$ and targets $t$ loss $L(y, t)$ is defined as follows:

$$L(y, t) = \frac{1}{B} \sum_i (\log(|y|) - \log(|t_i|))^2, \tag{6}$$

where $B$ is the batch size. Since $\log(|y|) = 1/B \sum \log(|t_i|)$ minimizes the MSE function, then the optimal value for $y$ equals to geometric mean $|y| = \exp(1/B \sum_{i=1}^N \log(|t_i|))$. Thus, the agent trained with our loss (6) will be optimized to predict the geometric mean of the expected return.

## 4.3 TREEDQN

The TreeMDP significantly differs from the usual temporal MDP process, so the standard RL algorithms would not work in this formulation. Thus, we proposed a computationally efficient TreeDQN method shown in Alg. 1.

---

**Algorithm 1** TreeDQN with experience replay

---

**Input:** Buffer size $N$, buffer min size $n$, discount factor $\gamma$, number of updates $t$, $\varepsilon$ decay function, batch size $b$, target update frequency $t_{up}$, random number generator R
**Initialize:** $Q_{\text{target}}, Q_{\text{net}}, \mathcal{D} \leftarrow \varnothing, \varepsilon \leftarrow 1$
**Result:** $Q_{\text{net}}$
$s \leftarrow$ env.reset()
**while** $i \leq t$ **do**
  **if** R(0, 1) $< \varepsilon$ **then**
    a $\leftarrow$ random action
  **else**
    a $\leftarrow \arg\max_a \left(-\exp\left(Q_{\text{net}}(s, a)\right)\right)$
  **end if**
  $s_{\text{next}}$, r, $s^+$, $s^-$, done = env.step(a)
  $\mathcal{D} \leftarrow \mathcal{D} \cup (s, a, r, s^{\pm})$
  **if** done **then**
    $s_{\text{next}} \leftarrow$ env.reset()
  **end if**
  $s \leftarrow s_{\text{next}}, \varepsilon \leftarrow$ decay$(\varepsilon), i \leftarrow i + 1$
  **if** $i > n$ **then**
    sample batch $(s, a, r, s^{\pm}) \sim \mathcal{D}$
    $a^{\pm} \leftarrow \arg\max_{a^{\pm}} \left(-\exp\left(Q_{\text{net}}(s^{\pm}, a^{\pm})\right)\right)$
    target $= r - \gamma \exp\left[Q_{\text{target}}(s^+, a^+)\right] -$
      $\gamma \exp\left[Q_{\text{target}}(s^-, a^-)\right]$
    loss $= \left(Q_{\text{net}}(s, a) - \log(|\text{target}|)\right)^2$
    $Q_{\text{net}} \leftarrow$ optimize$(Q_{\text{net}}, \text{loss})$
  **end if**
  **if** $i \mod t_{up} = 0$ **then**
    $Q_{\text{target}} \leftarrow Q_{\text{net}}$
  **end if**
**end while**

---

According to *Theorem 4.1*, the Bellman operator for a tree MDP process is contracting in mean. Hence, it can be used to learn a Q-function for the tree MDP process. Our method is inspired by the

Table 1: Loss functions used by RL algorithms to learn variable selection task.

| METHOD | LOSS FUNCTION |
| --- | --- |
| TREEDQN (OURS) | $L_{TreeDQN} = \left( Q(s_t, a_t) - \log \left( \left\| r - e^{Q_{target}(s_{t+1}^+, a_{t+1}^+)} - e^{Q_{target}(s_{t+1}^-, a_{t+1}^-)} \right\| \right) \right)^2$ |
| TMDP+DFS | $L_{tMDP+DFS} = -\log \pi_\theta(a_t|s_t) R(s_t) - \lambda H(\pi_\theta(\cdot|s_t))$ |
| FMSTS | $L_{FMSTS} = \left( \dfrac{Q(s_t, a_t) - R(s_t)}{size(root(s_t))} \right)^2$ |

Double Dueling DQN (Deep Q-Network) (Mnih et al., 2015) algorithm for a tree MDP process (2), but it has several key differences.

First, we operate with multiple next states denoted in the Algorithm by $s^+$, $s^-$. Thus, to compute the target for the Q-function, we predict the returns for all sub-trees. In Alg. 1, we store previous experiences in the form of $(s_t, a_t, r_t, s_{t+1}^+, s_{t+1}^-)$ in the buffer with capacity $N$. The number of stored next nodes varies since the node $s_t$ can have 0, 1, or 2 next nodes. The agent starts training when the buffer size reaches minimum capacity $n$; the agent starts training. In our implementation, we update the agent at each step of the environment using a batch of previous experiences from the replay buffer. This dramatically improves the sample efficiency compared to the on-policy method proposed for variable selection (Scavuzzo et al., 2022).

Second, the size of the tree can grow exponentially, we propose a different loss function that can be computed in a numerically stable way. We approximate a wide range of expected returns by exploiting a Graph Convolutional Neural Network with activation $f = -\exp(\cdot)$ applied to the output layer. Using logits before activation, we implement the MSLE loss (Eq. 6) in a numerically stable way for this activation function:

$$L_{TreeDQN} = \left( Q(s_t, a_t) - \log \left( \left\| r - e^{Q_{target}(s_{t+1}^+, a_{t+1}^+)} - e^{Q_{target}(s_{t+1}^-, a_{t+1}^-)} \right\| \right) \right)^2, \quad (7)$$

where $-e^{Q(s_t, a_t)}$ predicts the expected return for state-action pair $(s_t, a_t)$, $Q_{target}$ is the delayed version of $Q(s, a)$. The proposed loss function serves two purposes simultaneously: it optimizes the target value and geometric mean of expected return and stabilizes the learning process.

## 5 EXPERIMENTAL RESULTS

### 5.1 SYNTHETIC DATA

We compare the performance of our TreeDQN agent with the Strong Branching rule, Imitation Learning (IL) (Gasse et al., 2019), tMDP+DFS (on-policy, REINFORCE-based method) (Scavuzzo et al., 2022), and FMSTS (off-policy method) (Etheve et al., 2020) agents. We use the same Graph Convolutional Neural Network encoder architecture for all agents in our benchmarks. In addition, we present evaluation results for the SCIP solver with default parameters. However, it is not a direct competitor to our method since internal branching rules can make several modifications to the state of the solver except branching (Gamrath & Schubert, 2018).

The detailed comparison of our loss function with existing losses from tMDP+DFS ($L_{tMDP+DFS}$) and FMSTS ($L_{FMSTS}$) are provided in Table 1. Here, $R(s_t)$ is cumulative discounted reward, $H(\pi_\theta(\cdot|s_t))$ is the entropy of the policy $\pi_\theta(\cdot|s_t)$; $size(root(s))$ is the size of tree which contains node $s_t$; $Q_{target}$ is a delayed version of the state-action value function $Q$, actions in the next states $a_{t+1}^\pm$ are selected to maximize the expected return $a^\pm = \arg\max_{a^\pm} \left( -\exp\left( Q(s_{t+1}^\pm, a_{t+1}^\pm) \right) \right)$.

### 5.1.1 ENVIRONMENT

We use an open-source implementation of the B&B algorithm in SCIP solver with Ecole (Prouvost et al., 2020) v0.8.1 package, which represents nodes in the B&B tree as bipartite graphs and provides an interface for learning a variable selection policy.

**Observation.** The agent observes a bipartite graph. In this graph, edges correspond to connections between constraints and variables with weight equal to the coefficient of the variable in the constraint. Each variable and constraint node is represented by a vector of 19 and 5 features, respectively. The variables and constraints are represented by their properties, such as type (integer, binary, or continuous), upper and lower bounds, average incumbent value, dual bound value, and tightness of the constraint.

**Actions.** The agent selects one of the fractional variables for splitting. Since the number of fractional variables decreases during an episode, we apply a mask to choose only among available variables.

**Rewards.** At each step, the agent receives a negative reward $r = -1$. The total cumulative return equals the resulting tree size with a negative sign.

**Episode.** In each episode, the agent solves a single MILP instance. We limit the solving time for one task instance during training to 10 minutes and terminate the episode if the time is over.

### 5.1.2 TRAINING AND EVALUATION

We train our agent on a traditional set of NP-hard tasks, namely, Combinatorial Auction (Leyton-Brown et al., 2000), Set Cover (Balas & Ho, 1980), Maximum Independent Set (Bergman et al., 2016), Facility Location (Cornuejols et al., 1991) and Multiple Knapsack (Fukunaga, 2009). During training, we randomly generate MILP instances (e.g., capacities of knapsacks and values of items in the Multiple Knapsack task). Depending on sampled parameters, the task could be easy (LP relaxation of the initial problem provides an integer feasible solution, so the resulting B&B tree contains only the root node) or require multiple branching decisions to find an optimal solution. We use DFS for node selection for training and switch to SCIP default node selection policy for testing. For evaluation, we generate 40 task instances for each set of tasks and evaluate our agent with five random seeds. The parameters of the task distributions are shown in Appendix C, Tab. 5.

We use the same set of hyperparameters (Appendix C, Tab. 6) to train our agent for each task. The TreeDQN algorithm is robust to the hyperparameter choices. The number of training episodes was adjusted so the validation tree size converges. The epsilon decay parameter was adjusted to decay exploration to zero at the end of training. We seed the parameter $\gamma$, which defines the greediness of the agent, to 1 since it needs to minimize the tree size, and each tree node should have the same impact on the training. The rest of the hyperparameters have standard values from the literature. Total training time did not exceed four days on an Intel Xeon 6326, NVIDIA A100 machine. To select the best checkpoint for testing, we perform validation using 20 fixed task instances with five random seeds every 50 training episodes. The validation plot in Fig. 1 shows the geometric mean of tree sizes as a function of the number of training episodes. During training, the TreeDQN agent learns to solve variable selection tasks better, generating smaller B&B trees. As seen from Fig. 1, our off-policy TreeDQN method trains much faster than the on-policy tMDP+DFS method. The number of episodes it took to reach the best checkpoint and validation plots for all tasks is shown in Appendix C, Tab. 7, and Fig. 4.

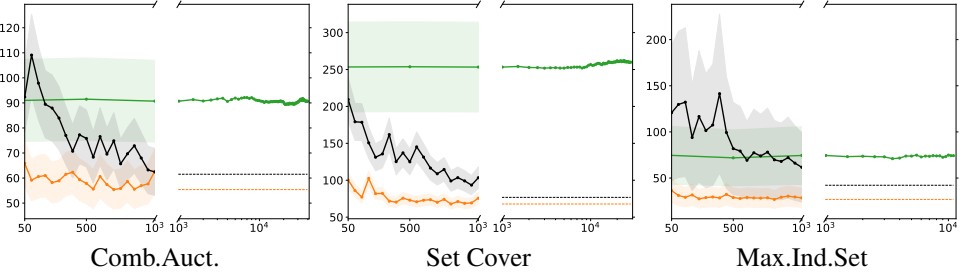

Figure 1: The geometric mean of tree size as a function of the number of training episodes. Orange - TreeDQN, black - FMSTS, green - tmdp+DFS. The dashed black and orange lines denote the best performance of the FMSTS and TreeDQN agents, respectively on the validation tasks trained on 10k episodes

To collect the dataset for the imitation learning method, we solve the MILP tasks with the B&B algorithm. At each step, we make a branching decision according to either the strong branching

heuristic with $p = 0.3$ or the pseudocost heuristic with $p = 0.7$ (see (Prouvost et al., 2020) for more details on the pseudocost heuristic). We add to the dataset only the branching decisions of the strong branching heuristic. The pseudocost heuristic acts as an exploration policy to provide more diversity to the dataset. For all tasks, we used 83000 data points for training and 17000 for validation to prevent overfitting.

In Tab. 2, we show the geometric mean of execution time and standard deviation. Bold numbers indicate the best-performing RL method (TreeDQN, FMSTS, tmdp+DFS). All RL methods perform much faster than the Strong branching, with execution time proportional to the number of nodes in the B&B trees (see Appendix D, Tab. 8). The SCIP default and the Strong branching have higher execution times when compared to the learning-based methods due to the computational complexity of their branching rules. Additional analysis of tree size distributions and running time is available at Appendix D.

To prove the statistical significance of our results, we perform paired difference test (Wilcoxon, 1945) between our method and baselines. Our null hypothesis is that TreeDQN performs similarly to our baselines, so the distribution of differences in execution times should be symmetric about zero. The results, shown at the bottom of Tab. 2, indicate strong evidence against the null hypothesis for almost all test tasks except Facility Location, where TreeDQN performs close to IL and Multiple Knapsack where RL methods (TreeDQN, FMSTS, and tmdp+DFS) perform close to each other.

Table 2: Geometric mean of execution time. TreeDQN(MSE) is a TreeDQN agent trained with MSE loss function. W() denotes the Wilcoxon test between TreeDQN and methods in brackets.

| Model | Comb. Auct. | Set Cover | Max. Ind. Set | Facility Loc. | Mult. Knap. |
|---|---|---|---|---|---|
| SCIP default | $2.01 \pm 1.58$ | $2.87 \pm 1.58$ | $3.69 \pm 2.08$ | $7.27 \pm 2.54$ | $0.52 \pm 1.94$ |
| Strong Branching | $3.36 \pm 2.21$ | $5.33 \pm 2.22$ | $57.03 \pm 3.48$ | $24.81 \pm 4.85$ | $1.94 \pm 4.76$ |
| IL | $0.77 \pm 1.49$ | $1.07 \pm 1.50$ | $1.75 \pm 1.65$ | $3.99 \pm 2.92$ | $1.83 \pm 4.42$ |
| TreeDQN (ours) | $\mathbf{0.79 \pm 1.53}$ | $\mathbf{1.09 \pm 1.52}$ | $\mathbf{1.86 \pm 1.88}$ | $\mathbf{3.90 \pm 2.84}$ | $\mathbf{0.90 \pm 2.75}$ |
| TreeDQN (MSE) | $0.81 \pm 1.54$ | $1.18 \pm 1.55$ | $1.97 \pm 1.74$ | $4.60 \pm 3.02$ | $1.11 \pm 2.93$ |
| FMSTS | $0.82 \pm 1.58$ | $1.21 \pm 1.63$ | $2.45 \pm 2.41$ | $5.39 \pm 3.49$ | $0.93 \pm 2.95$ |
| tmdp+DFS | $0.90 \pm 1.64$ | $1.69 \pm 2.01$ | $1.96 \pm 1.61$ | $5.29 \pm 3.62$ | $0.90 \pm 2.89$ |
| W(IL) | $2.37 \cdot 10^{-9}$ | $1.08 \cdot 10^{-3}$ | $2.68 \cdot 10^{-4}$ | $2.66 \cdot 10^{-1}$ | $4.38 \cdot 10^{-9}$ |
| W(MSE) | $1.86 \cdot 10^{-3}$ | $6.88 \cdot 10^{-29}$ | $1.26 \cdot 10^{-8}$ | $4.17 \cdot 10^{-6}$ | $4.31 \cdot 10^{-2}$ |
| W(FMSTS) | $7.47 \cdot 10^{-5}$ | $1.61 \cdot 10^{-27}$ | $2.16 \cdot 10^{-26}$ | $1.44 \cdot 10^{-14}$ | $9.77 \cdot 10^{-1}$ |
| W(tmdp+DFS) | $1.34 \cdot 10^{-27}$ | $2.39 \cdot 10^{-34}$ | $5.00 \cdot 10^{-9}$ | $1.40 \cdot 10^{-13}$ | $5.28 \cdot 10^{-1}$ |

We compared the performance of our agent and the TreeDQN agent trained with standard MSE loss function. Our modified learning objective prevents explosions of gradients and significantly stabilizes the training process. Training with smoother gradients should lead to a better policy to solve the MILP tasks faster. As seen from Tab. 2 in all tasks, the agent trained with a modified loss function achieves a lower geometric mean of the execution time. The Wilcoxon test (Wilcoxon, 1945) indicates statistical significance that our modified loss function allows our agent to learn a better policy for all tasks. Additionally, we tested the generalization ability of our agent and evaluated the trained agent on the large instances from the transfer distribution. Refer to the Appendix E for details.

The experimental results on synthetic tasks demonstrate that our TreeDQN agent outperforms baseline RL methods. However, strong branching is often close to the optimal policy on simple tasks. Consequently, the IL will be a strong baseline for such tasks. In the next section, we demonstrate the performance of our method on more complex tasks with different maximization objectives, where imitation learning is not as close to optimal as in synthetic tasks.

## 5.2 Real data

We evaluate our method on a challenging Balanced Item Placement dataset (ML4CO competition Gasse et al. (2022), BSD-3-Clause license). The latter focuses on a data-driven design of

application-specific branching algorithms. The balanced item placement problem is computationally demanding and requires efficient sample algorithms to learn a branching policy. In the dataset, problem instances are modeled as multi-dimensional multi-knapsack MILP tasks. Each task represents the spread of items across containers, e.g., files across disks or distributing processes across different machines with even utilization. The number of movable items is constrained to model the real-life situation of a live system. The dataset contains 9900 train instances, 100 validation, and 100 test instances.

### 5.2.1 ENVIRONMENT

**Observations and actions.** The agent observes a bipartite graph and returns an index of a variable for splitting.

**Rewards.** We train the agent to maximize the dual integral. The dual integral measures the area under the curve of the solver's global lower bound (dual bound), corresponding to a solution of LP relaxation of the MILP. When the agent chooses branching variables, the domain of integer variables gets tightened, and the dual bound increases over time. The dual integral is defined as follows:

$$I_d = \int_{t=0}^{T} z_t^* dt,$$

where $T$ is the time limit, and $z_t^*$ is the best dual bound found at time $t$. At each time step, the agent receives the reward equal to the dual integral since the previous state, so the cumulative return equals the dual integral $I_d$.

Note the different reward functions here. In the synthetic experiments, we use the final tree size as our reward since the tasks are sufficiently simple and can be solved conveniently. Thus, the agent trained to minimize the tree size could solve the underlying MILP faster since it would require fewer branching decisions. However, the ML4CO tasks are much harder, and the optimal solutions can not be obtained in a reasonable time. So, we use a different reward metric that does not rely on solving the task optimally.

**Episode.** In each episode, the agent solves a single MILP instance. The episode duration is limited to 15 minutes during both training and evaluation.

### 5.2.2 TRAINING AND EVALUATION

We train our agents with the same hyperparameters and the same architecture as in synthetic tasks. Since each episode in this environment takes 15 minutes to complete, we decrease the number of training episodes to 500. This environment highlights the sample efficiency of our method because training on policy is computationally complex.

Table 3: Evaluation on balanced item placement task.

| MODEL | REWARD | PRIMAL BOUND | DUAL BOUND | # NODES $\times 10^3$ | # LPs $\times 10^3$ |
|---|---|---|---|---|---|
| SCIP DEFAULT | 3885.24 | 18.46 | 4.97 | 258.36 | 5037.10 |
| STRONG BRANCHING | 3419.00 | 628.02 | 4.01 | 0.552 | 13.95 |
| IL | 4964.77 | 537.85 | 5.92 | 141.36 | 1911.16 |
| TREEDQN (OURS) | **5958.06** | **87.33** | **7.05** | 83.76 | 846.40 |

We compare the performance of the TreeDQN agent with the SCIP solver, Strong Branching heuristic, and Imitation Learning agent on 100 test instances. Since the tasks are computationally demanding, all tasks for each branching method were finished by reaching the 15-minute time limit. We present evaluation results in Tab. 3. Here, the TreeDQN agent achieves the highest cumulative reward by a significant margin. Comparing the TreeDQN and IL agents, which use the same GCNN architecture, we see that for the same amount of time, TreeDQN solves significantly fewer LP tasks. This is because it creates more complex LPs, which increase the dual bound faster.

Fig. 2 shows the primal and dual bounds as a function of time. Our TreeDQN agent decreases the primal bound and increases the dual bound much faster than the IL and Strong Branching agents.

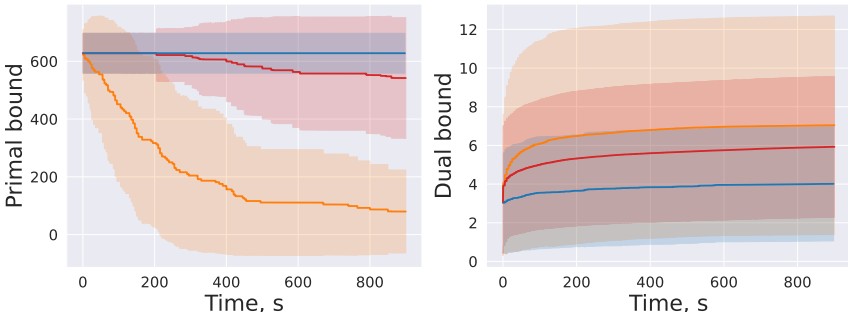

Figure 2: Primal bound (on the top) and dual bound (on the bottom) as a function of time. Red - Imitation Learning, orange - TreeDQN, blue - Strong Branching.

Table 4: Dual bound distribution.

| MODEL | 25% | 50% | 75% | 100% |
|---|---|---|---|---|
| SCIP DEFAULT | 1.18 | 3.76 | 7.24 | 20.88 |
| STRONG BRANCHING | 1.72 | 3.54 | 5.88 | 12.95 |
| IL | 3.65 | 5.18 | 7.93 | 17.53 |
| TREEDQN (OURS) | 3.37 | 5.59 | 9.37 | 23.27 |

We present quantiles of the dual bound distributions in Tab. 4. Here, the dual bound distribution of the TreeDQN agent has larger 50%, 75%, and 100% quantiles than the distributions of other methods. These results demonstrate that our agent learns an effective branching policy, is sample-efficient, and can be trained with only 500 training episodes.

## 6  CONCLUSION

In this paper, we have presented a novel data-efficient deep RL method to learn a branching rule for the B&B algorithm (Alg. 1) and demonstrated its superiority over the existing RL-based techniques (Tab. 2, 3). We provided statistical tests (Tab. 2) which confirm the statistical significance of our experimental results. The synergy of the exact solving algorithm and data-driven heuristic takes advantage of both worlds: guarantees the computation of the optimal solution from the B&B algorithm and the ability to adapt to specific tasks from the learned branching heuristic. In particular, we introduced the novel loss function (7), which stabilizes the training process in the presence of high-variance returns and proves its superiority over the alternative approaches listed in Tab. 1. As a typical RL application to MILP, our method is designed to perform well on the distribution of similar MILP tasks (see formal task definition in Section 2). Thus, transferring the trained policy to significantly different MILP tasks is out of the scope of the present paper and is considered a promising direction for future research. We experimentally demonstrated the high performance of our method on a set of synthetic and practical tasks compared to previously known solutions. Our TreeDQN method trains much faster than the previous RL methods, as shown in Fig. 1, it outperforms state-of-the-art RL methods as demonstrated in Tab. 2 and can learn an efficient branching policy which outperforms imitation learning method in complex practical task when trained only on 500 episodes which would not be possible with sample inefficient on-policy methods. The source code is available at `https://anonymous.4open.science/r/treedqn-F72F`.

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

## A    BRANCH-AND-BOUND ALGORITHM

---

**Algorithm 2** Branch-and-Bound

---

Set the original MILP problem as the root of the tree
Set GUB $= +\infty$
**while** not solved **do**
    1. Select a not visited node from the tree using *node selection strategy*. Compute the LB as a solution of the relaxed problem: $\min(c^T \hat{x} : A\hat{x} \leq b, \hat{x} \in [l, u], \hat{x} \in \mathbb{R}^n)$
    2. Update the GUB if the relaxed problem provides a feasible solution
    3. If LB < GUB and the corresponding MILP is feasible, choose one of the fractional variables $\hat{x}_i$ using the *branching rule*, split its domain into two parts with constraints $l_i \leq x_i \leq \lfloor \hat{x}_i \rfloor$ and $\lceil \hat{x}_i \rceil \leq x_i \leq u_i$, and produce two corresponding nodes as descendants of the visited nodes in the tree
    4. Mark the selected node as visited
**end while**

---

## B    DISTRIBUTION OF TREE SIZES

We present the distribution of tree sizes obtained with the Strong branching heuristic in Fig. 3. It is seen that even the Strong branching heuristic produces a long-tailed distribution of tree sizes.

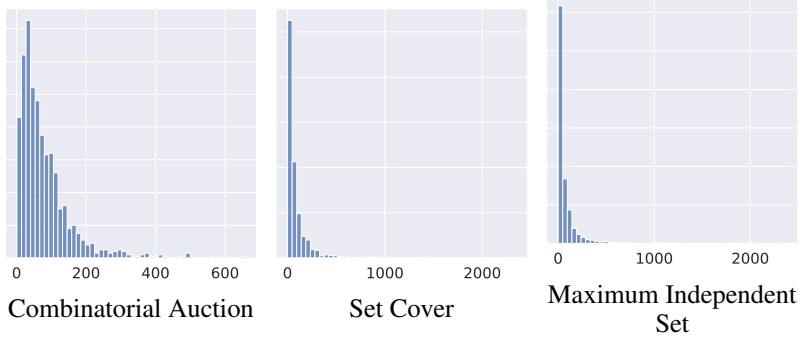

Figure 3: Distributions of tree sizes for Combinatorial Auction (Leyton-Brown et al., 2000), Set Cover (Balas & Ho, 1980) tasks and Maximum Independent Set (Bergman et al., 2016) using Strong Branching heuristic for variable selection.

## C    TRAINING PARAMETERS

Tab. 5 shows the parameters used to generate synthetic tasks. The train, validation, and test tasks share the same number of variables and constraints. Transfer tasks have a more significant number of variables and constraints. Tab. 6 shows hyperparameters of our TreeDQN agent. The number of training episodes was 1000 for synthetic tasks and 500 for the Balanced Item Placement task.

Table 5: Parameters used to generate train, validation, test, and transfer tasks. Combinatorial Auction (items/bids), Set Cover(rows/cols), Maximum Independent Set(nodes), Facility Location (clusters/facilities), Multiple Knapsack (items/knapsacks).

|  | COMB. AUCT. | SET COVER | MAX. IND. SET | FACILITY LOC. | MULT. KNAP. |
|---|---|---|---|---|---|
| TEST | 100 / 500 | 400 / 750 | 500 | 35 / 35 | 100 / 6 |
| TRANSFER | 200 / 1000 | 500 / 1000 | 1000 | 60 / 35 | 100 / 12 |

Fig. 4 shows the geometric mean of the tree size as a function of number of training episodes for a fixed 20 validation task instances evaluated with five random seeds. Tab. 7 shows the number of

Table 6: Hyperparameters used in the training of the TreeDQN agent.

| PARAMETER | VALUE |
|---|---|
| $\gamma$ | 1 |
| BUFFER SIZE | 100'000 |
| BUFFER MIN SIZE | 1'000 |
| BATCH SIZE | 32 |
| LEARNING RATE | $10^{-4}$ |
| $\varepsilon$-DECAY STEPS | 100'000 |
| NUMBER OF TRAINING EPISODES | 1000 |
| OPTIMIZER | ADAM |

episodes required to reach the smallest geometric mean on validation tasks for the RL methods used in our benchmarks.

Table 7: Number of training episodes required to reach the best checkpoint.

| MODEL | COMB. AUCT. | SET COVER | MAX. IND. SET | FACILITY LOC. | MULT. KNAP. |
|---|---|---|---|---|---|
| TREEDQN | 700 | 800 | 800 | 200 | 850 |
| FMSTS | 1000 | 950 | 50 | 200 | 400 |
| TMDP+DFS | 22500 | 3000 | 3500 | 6500 | 9500 |

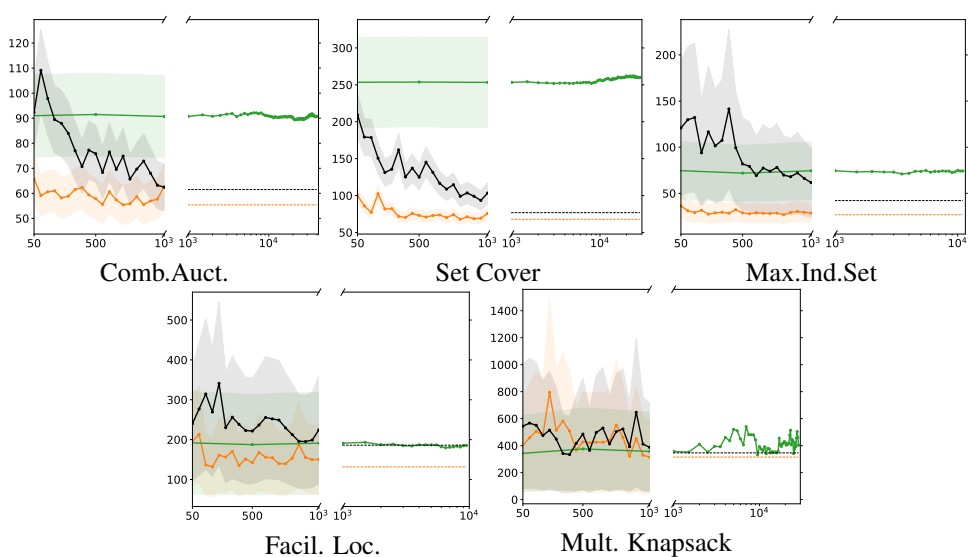

Figure 4: The geometric mean of tree size as a function of the number of training episodes. Orange - TreeDQN, black - FMSTS, green - tmdp+DFS.

## D    TEST TASKS

### D.1    GEOMETRIC MEAN OF TREE SIZES

In Tab. 8, we show the geometric mean of the final tree size and geometric standard deviation. Tab. 8 shows that the TreeDQN agent significantly exceeds the results of the tmdp+DFS and FMSTS agents in all test tasks. The TreeDQN agent is close to the Imitation Learning agent in the first four tasks and substantially outperforms the Imitation Learning and Strong Branching in the Multiple Knapsack tasks.

Table 8: Geometric mean of tree size with geometric std for test tasks.

| MODEL | COMB. AUCT. | SET COVER | MAX. IND. SET | FACILITY LOC. | MULT. KNAP. |
|---|---|---|---|---|---|
| TREEDQN (OURS) | **58 ± 3** | **56 ± 2** | **42 ± 6** | **324 ± 8** | **290 ± 6** |
| TREEDQN (MSE) | 62 ± 3 | 63 ± 2 | 60 ± 5 | 398 ± 9 | 358 ± 7 |
| FMSTS | 65 ± 3 | 76 ± 3 | 96 ± 8 | 499 ± 10 | 299 ± 6 |
| TMDP+DFS | 93 ± 3 | 204 ± 3 | 88 ± 4 | 521 ± 10 | 308 ± 6 |

An important metric is the gap between primal and dual bounds as a function of time shown in Fig. 5. In the B&B algorithm, the primal-dual gap monotonically decreases when solving a task instance. The speed of the gap reduction is proportional to the number of nodes and mean execution time.

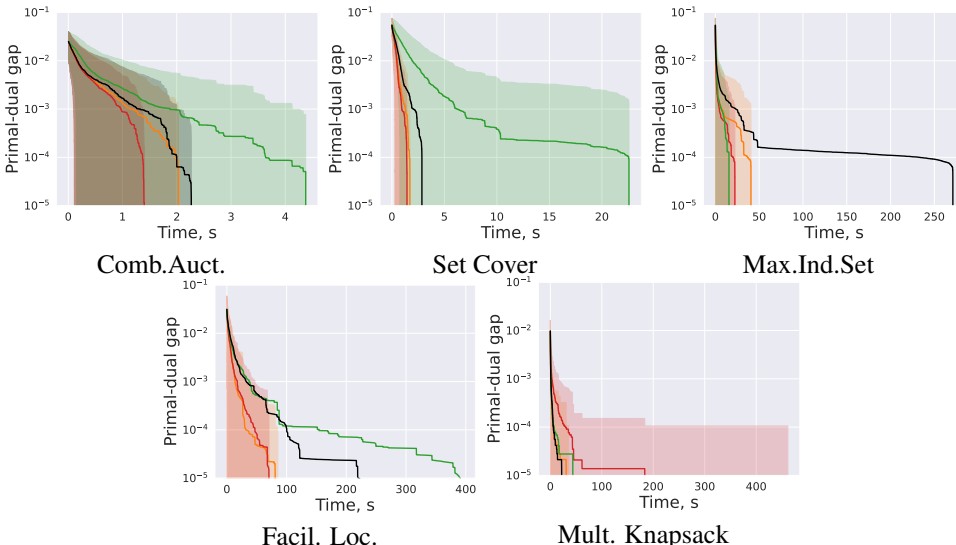

Figure 5: Primal-dual gap as a function of time. Red - IL, orange - TreeDQN, black - FMSTS, green - tmdp+DFS.

### D.2 PROBABILITY-PROBABILITY PLOTS

To further analyze the performance of our agent, we present distributions of tree sizes in the form of probability-probability plots (P-P plots) in Fig. 6. P-P plot allows us to compare different cumulative distribution functions (CDF). For a reference CDF $F$ and a target CDF $G$ P-P plot is constructed similar to the ROC curve: we choose a threshold $x$, move it along the domain of $F$ and draw points $(F(x), G(x))$. To show multiple distributions on the same plot, we use Strong Branching as reference CDF for all of them. If one curve is higher than another, the corresponding CDF is more extensive so that the associated agent can solve more tasks in $x$ or nodes or less. All our baselines (except Strong Branching) have close complexity per call. So, if one curve is higher than another, the corresponding agent can solve more tasks at the same time. This is related to winning rates, which shows the number of instances solved in a specific time limit (see Appendix D.3, Tab. 9, 10, 9). From the P-P plot for the Maximum Independent Set, we see that TreeDQN is good at solving simple tasks where it performs better than Imitation Learning. When the tasks become more complex, the performance of TreeDQN decreases. This behavior is the direct consequence of our learning objective. We optimize the geometric mean of expected tree size, so complex task instances may have less influence on the learning process.

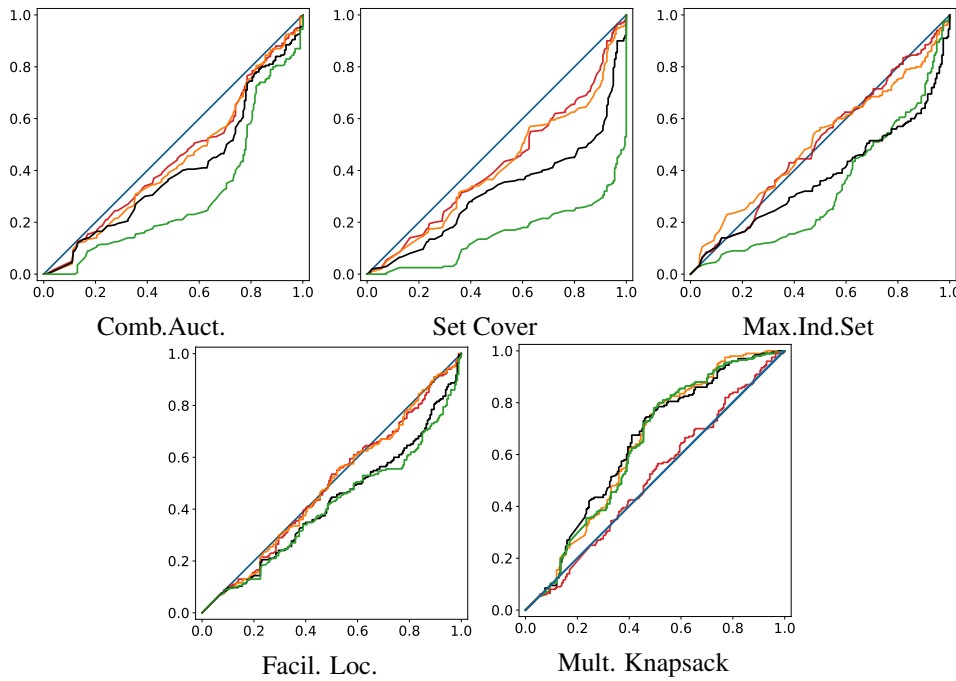

Figure 6: P-P plots of tree size distributions for test instances. Blue - Strong Branching, red - Imitation Learning, orange - TreeDQN, black - FMSTS, green - tmdp+DFS.

### D.3 WINNING RATES

We show winning rates for Combinatorial Auction and Set Cover tasks in Tab. 9, for Maximum Independent Set and Facility Location tasks in Tab. 10, and for Multiple Knapsack task in Tab. 11. The TreeDQN agent performs close to the IL agent in the first four tasks and outperforms it in the Multiple Knapsack task.

Table 9: Number of instances solved to optimal solution simultaneously as IL agent solves 25%, 50%, 75%, and 100% of Combinatorial Auction and Set Cover tasks.

| | COMB.AUCT | | | | SET COVER | | | |
|---|---|---|---|---|---|---|---|---|
| MODEL | 25% | 50% | 75% | 100% | 25% | 50% | 75% | 100% |
| SCIP DEFAULT | 0.00% | 0.00% | 0.00% | 22.00% | 0.00% | 0.00% | 0.00% | 7.50% |
| STRONG BRANCHING | 0.00% | 0.00% | 0.00% | 20.00% | 0.00% | 1.50% | 2.00% | 12.50% |
| IL | 25.00% | 50.00% | 75.00% | 100.00% | 25.00% | 50.00% | 75.00% | 100.00% |
| TREEDQN | 21.50% | 47.50% | 73.50% | 96.50% | 21.00% | 53.00% | 72.50% | 97.50% |
| FMSTS | 23.00% | 43.50% | 70.00% | 94.50% | 17.50% | 36.50% | 51.50% | 93.00% |
| TMDP+DFS | 16.50% | 28.50% | 61.50% | 92.50% | 6.50% | 23.50% | 35.00% | 69.00% |

Table 10: Number of instances solved to optimal solution simultaneously as IL agent solves 25%, 50%, 75%, and 100% of Maximum Independent Set and Facility Location tasks.

| Model | Max.Ind.Set | | | | Facility Loc. | | | |
|---|---|---|---|---|---|---|---|---|
| | 25% | 50% | 75% | 100% | 25% | 50% | 75% | 100% |
| SCIP DEFAULT | 0.00% | 0.00% | 0.00% | 99.00% | 7.00% | 16.00% | 42.50% | 100.00% |
| STRONG BRANCHING | 0.00% | 0.00% | 0.00% | 11.86% | 2.15% | 4.84% | 18.82% | 69.89% |
| TREEDQN | 29.38% | 53.09% | 71.65% | 97.42% | 24.73% | 48.92% | 79.57% | 99.46% |
| FMSTS | 21.13% | 32.99% | 54.12% | 95.36% | 20.43% | 38.71% | 60.22% | 97.85% |
| TMDP+DFS | 10.82% | 31.44% | 57.22% | 100.00% | 20.97% | 43.55% | 61.29% | 97.85% |

Table 11: Number of instances solved to optimal solution simultaneously as IL agent solves 25%, 50%, 75%, and 100% of Multiple Knapsack tasks.

| Model | 25% | 50% | 75% | 100% |
|---|---|---|---|---|
| SCIP DEFAULT | 43.00% | 94.00% | 100.00% | 100.00% |
| STRONG BRANCHING | 16.93% | 48.15% | 74.60% | 99.47% |
| TREEDQN | 42.33% | 76.72% | 93.12% | 100.00% |
| FMSTS | 44.97% | 76.19% | 91.53% | 100.00% |
| TMDP+DFS | 34.39% | 76.19% | 93.12% | 100.00% |

# E    TRANSFER TASKS

Besides testing the performance of our agent, we also study its abilities to generalize. Tables 12 and 13 present evaluation results for complex transfer tasks solved with five different seeds. Since solving complicated MILP problems is time-consuming, we limit the maximum number of nodes in a B&B tree to $200'000$. The number of transfer tasks terminated by this node limit is shown in Tab. 14. We use node limit as tree size for terminated instances when computing the geometric mean. It is seen from Tab. 12 that in the Set Cover, Facility Location, and Multiple Knapsack tasks, our TreeDQN agent transfers well and performs better than the tmdp+DFS and FMSTS agents. In the Combinatorial Auction task, the FMSTS agent transfers slightly better than TreeDQN. In the Maximum Independent Set task, the TreeDQN agent falls behind the tmdp+DFS agent since it adapted better for simple task instances.

Table 12: Geometric mean of execution time for transfer tasks.

| Model | Comb.Auct | Set Cover | Max.Ind.Set | Facility Loc. | Mult.Knap. |
|---|---|---|---|---|---|
| SCIP DEFAULT | $42.47 \pm 1.81$ | $13.34 \pm 1.93$ | $116.29 \pm 2.50$ | $34.33 \pm 3.78$ | $44.29 \pm 2.66$ |
| IL | $19.93 \pm 2.24$ | $8.89 \pm 2.17$ | $171.89 \pm 6.64$ | $43.67 \pm 6.83$ | $514 \pm 5.26$ |
| TREEDQN | $42.01 \pm 3.42$ | $\mathbf{9.86 \pm 2.34}$ | $204.99 \pm 5.76$ | $\mathbf{52.84 \pm 3.85}$ | $\mathbf{302.43 \pm 4.91}$ |
| FMSTS | $\mathbf{30.01 \pm 2.66}$ | $13.28 \pm 2.74$ | $417.27 \pm 7.31$ | $82.43 \pm 4.21$ | $372.56 \pm 5.12$ |
| TMDP+DFS | $45.88 \pm 2.96$ | $42.33 \pm 4.12$ | $\mathbf{68.39 \pm 3.55}$ | $66.90 \pm 3.22$ | $358.44 \pm 5.65$ |
| W(IL) | $6.33 \cdot 10^{-34}$ | $6.71 \cdot 10^{-21}$ | $4.03 \cdot 10^{-1}$ | $8.09 \cdot 10^{-3}$ | $1.78 \cdot 10^{-7}$ |
| W(FMSTS) | $1.01 \cdot 10^{-17}$ | $5.87 \cdot 10^{-31}$ | $2.54 \cdot 10^{-13}$ | $9.89 \cdot 10^{-20}$ | $4.99 \cdot 10^{-1}$ |
| W(TMDP+DFS) | $5.67 \cdot 10^{-3}$ | $1.91 \cdot 10^{-34}$ | $1.89 \cdot 10^{-9}$ | $7.34 \cdot 10^{-1}$ | $3.22 \cdot 10^{-4}$ |

Tab. 14 shows the number of task instances finished by the node limit. It can be used to assess the worst-case performance of the trained RL algorithms when transferred from test to transfer distribution. In the Maximum Independent Set task, the FMSTS agent has the worst performance.

Table 13: Geometric mean of tree size with geometric std for transfer tasks.

| MODEL | COMB.AUCT | SET COVER | MAX.IND.SET | FACILITY LOC. | MULT.KNAP. |
|---|---|---|---|---|---|
| TREEDQN (OURS) | $1567 \pm 4$ | $\mathbf{174 \pm 4}$ | $4541 \pm 9$ | $\mathbf{759 \pm 11}$ | $\mathbf{35599 \pm 4}$ |
| FMSTS | $\mathbf{1375 \pm 3}$ | $252 \pm 4$ | $8647 \pm 9$ | $1135 \pm 11$ | $42461 \pm 5$ |
| TMDP+DFS | $2171 \pm 4$ | $858 \pm 6$ | $\mathbf{1713 \pm 5}$ | $847 \pm 10$ | $40316 \pm 5$ |

In the Multiple Knapsack task, the TreeDQN agent transfers better than the other trained algorithms, including imitation learning.

Table 14: Number of transfer tasks finished by node limit. The total number of instances with different seeds is 200.

| MODEL | COMB. AUCT. | SET COVER | MAX. IND. SET | FACILITY LOC. | MULT. KNAP. |
|---|---|---|---|---|---|
| SCIP DEFAULT | 0 | 0 | 0 | 0 | 2 |
| STRONG BRANCHING | 0 | 0 | 0 | 10 | 42 |
| IL | 0 | 0 | 5 | 0 | 44 |
| TREEDQN | 0 | 0 | 4 | 6 | 23 |
| FMSTS | 0 | 0 | 25 | 1 | 37 |
| TMDP+DFS | 0 | 0 | 1 | 2 | 36 |

## F LEARNING CURVES FOR TREEDQN

Fig. 7 shows the geometric mean of the tree size on validation task instances for TreeDQN (orange), SCIP default (red), and strong branching (black) methods as a function of the number of training episodes. As seen from Fig. 7, the variable selection policy of the TreeDQN agent improves during the training.

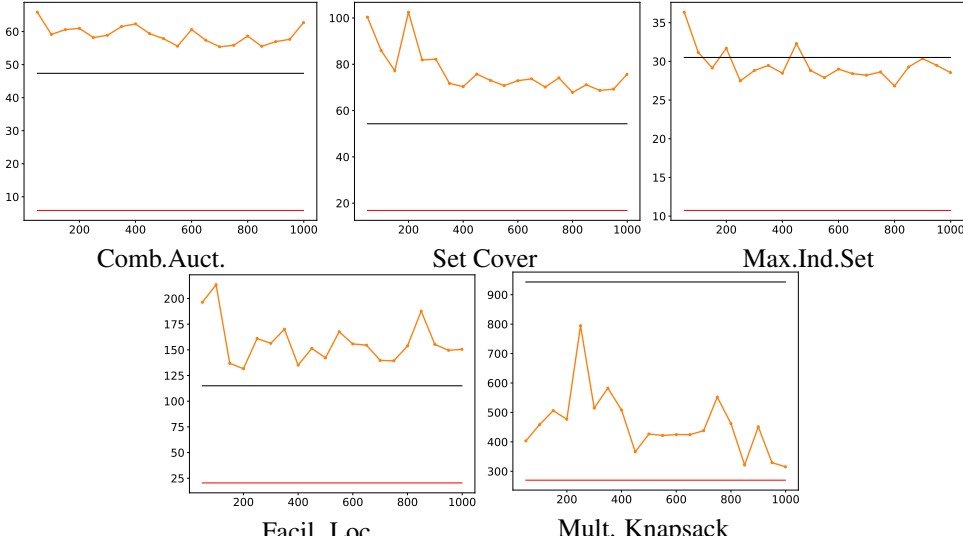

Figure 7: Geometric mean of tree size for as a function of the number of training episodes. Orange - TreeDQN, black - strong branching, red - SCIP default.

