# OpenReview forum: "TreeDQN: Sample-Efficient Off-Policy Reinforcement Learning for Combinatorial Optimization"
_ICLR.cc/2025/Conference — ICLR 2025 Conference Withdrawn Submission_

### Official Review · Reviewer_VNaC · 2024-10-27

**Soundness:** 2
**Presentation:** 3
**Contribution:** 2
**Rating:** 5
**Confidence:** 4

**Summary:**

Authors propose an off-policy RL algorithm, TreeDQN, to scale the branch and bound algorithm for combinatorial optimization. The branching nodes are infered with a neural net forward pass. This approach is claimed to be faster than classical solvers such SCIP and CPLX because search trees are smaller with the proposed TreeDQN. To me, the contribution compared to previous work that also apply RL to branch and bound is not clear.

**Strengths:**

The use of RL for B&B algorithm is well-motivated (second paragraph of the intro). The experimental setup is detailed.

**Weaknesses:**

I believe the contriubtions are unclear.

What is the difference between FMCTS and TreeDQN from a convergence point of view ? It seems to me that when you say on line 163 " This method [FMCTS] is sample efficient since training data can be sampled from a buffer of past experiences. However, it may not converge to the optimal policy because its training data was obtained by older and less efficient versions of the Q-function ", this aso applies to TreeDQN despite proving contraction of operators.

Furthermore, I think Tree MDPs can be rewritten as classical MDPs and thus one can just use the classical Bellman Operators to do Q-learning. Can't you just say your state space the set of all sub-MILPs coupled with the current tree depth, your actions are the set of branching node and the reward is -1 ? I am not sure why one needs to define tree MDPs and prove contractions of tree operators.

I am very troubled by figure 1. In particular, I am not convinced that TreeDQN actually learns anything: it seems to me that solutions found at initialization of TreeDQN are already better than those of FMCTS.

Furthermore, why can't you add tree sizes of solvers such as SCIP or CPLEX on your figure 1 ? This would be easier to compare your results with those of FMCTS (figure 2 of Etheve 2020).

You should add parenthesis to your citations in your introduction please (e.g. lines 50-54).

**Questions:**

Is it really necessary to define and prove results about a new Bellman Operator?

What is the difference between FMCTS and TreeDQN aside from the loss to optmize?

Why not include the tree sizes of SCIP or CPLX in your figure 1?

Are you sure TreeDQN is actually learning?

Thank you in advance

---

> ### Author Response · Authors · 2024-11-20
> **Thank you for the review.**
>
> First, we would like to mention that our method is off-policy RL, not offline RL, as you mentioned.  The offline RL learns from the offline data, i.e., it does not interact with the environment and tries to learn an optimal policy only from a fixed dataset. The off-policy RL interacts with the environment, it stores all the acquired data in the replay buffer and uses it to learn a policy. In contrast, on-policy RL also requires interaction with the environment but does not store the interaction data and is much more sample inefficient when compared to the off-policy RL methods.
>
> > Weakness 1
>
> Thank you for bringing it up. As the Related Work section mentions, the FMCTS method regresses the Q-function to the sampled return (i.e., cumulative discounted reward) obtained in one episode. The TreeDQN method uses the Bellman operator, so we do not need to store the policy-dependent sampled returns; we just store the rewards that do not depend on the policy. The implementation details are available in our Alg. 1 (lines 258 - 262).
>
> > Weakness 2
>
> Unfortunately, this is not possible directly since, in the Tree MDP, each node can have multiple child nodes. If we define the state space as the set of all sub-MILPs, the state space will become extremely large. The idea of reinforcement learning is similar to dynamic programming when you can decompose the task into smaller subtasks.
>
> > Weakness 3
>
> We apologize for the inconvenience. Fig. 1 starts with 50 updates of the neural network weights, as denoted on the X-axis. Before the first update, FMCTS and TreeDQN, with similar neural network initialization, will produce similar variable selection heuristics and, hence, similar B&B trees.
>
>
> > Weakness 4
>
> Fig. 1 illustrates the tree size on the evaluation dataset during the training. We do not plot the SCIP method here since we want to present the convergence speed of different reinforcement learning methods.
>
>
> > Weakness 5
>
> Thank you for noticing that! We fixed this issue.
>
>
> > Question 1
>
> We prove the results about the Bellman operator as theoretical motivation for our TreeDQN method. Due to the contraction property, we can use the Bellman operator to learn a Q-function in our TreeDQN method. Learning the Q-function with the Bellman operator is more theoretically correct than regressing the Q-function to cumulative returns obtained by older versions of the variable selection policy.
>
>
> > Question 2
>
> The TreeDQN, FMCTS, and tmdp+DFS use similar neural network architecture (see Section 5, lines 307 - 310). All three methods differ in how the training data is acquired and used to train the neural network.
> FMCTS stores the sampled returns (cumulative discounted rewards) and uses them to regress the Q-function (see sec. 3.3 lines 160 - 164).
> tmdp+DFS only uses the most recent integration data to perform a gradient update using the REINFORCE-based method (see sec. 3.3 lines 164 - 174).
> TreeDQN stores the rewards (not sampled returns) and uses the Bellman operator to learn the Q-function (see sec. 4).
>
>
> > Question 3
>
> See the answer to the weakness above.
>
>
> > Question 4
>
> See the answer to the weakness above.

---

> ### Comment · Reviewer_VNaC · 2024-11-20
> **Thank you but lower score**
>
> Dear Authors, thank you for answering some of my questions.
>
> Indeed, It is a typo I understood that you were using DQN which is off-policy. Apologies for this.
>
> Unfortunately, I am not convinced about your rebuttal.
>
> ```
> Unfortunately, this is not possible directly since, in the Tree MDP, each node can have multiple child nodes. If we define the state space as the set of all sub-MILPs, the state space will become extremely large. The idea of reinforcement learning is similar to dynamic programming when you can decompose the task into smaller subtasks.
> ```
> Thank you, for explaining what is dynamic programming and what is reinforcement learning. To me it is not a problem to define this state-space of all sub-MILPs with set notation. Then from a practical perspective you will not store all the possible states in memory but you will just implement the transition kernel of the MDP (the ```step()``` function in the ```gymnasium``` framework).
>
> ```
> We do not plot the SCIP method here since we want to present the convergence speed of different reinforcement learning methods.
> ```
>
> Isn't it exactly what Etheve et. al. 2020 do in their Figure 2? I find the fact that you refuse to add number of nodes of SCIP or CPLX on your plots suspicious. I also noticed that you use 'FMCTS' everywhere in your text to refer to Etheve et. al. 2020 but the actual name of the algorithm is 'FMSTS' if I am not mistaken
>
> You did not answer the key question: are you sure TreeDQN is actually learning? As I mentioned in my review, the learning curves of TreeDQN are very flat. Could you produce a plot isolating its learning curve (do Figure 1 with only TreeDQN) please?
>
> In the current state I cannot recommend acceptance of your work. I lowered my score until you convinced me that TreeDQN can actually learn B&B trees that are close in size to CPLX' or SCIP'.
>
> Thank you

---

> > ### Author Response · Authors · 2024-11-21
> > **Additional details**
> >
> > Thank you for the quick response. To address the concerns about the learning curves of our TreeDQN method, we added Appendix F. It shows the geometric mean of the tree size as a function of the number of training episodes. We also added geometric means of the tree sizes for the SCIP default and strong branching methods. We hope that this evaluation resolves your concerns.
> >
> > We would be glad to provide additional evaluations if needed. Thank you for spotting the typo in the name of the FMSTS method.

---

> > > ### Comment · Reviewer_VNaC · 2024-11-21
> > > **Thank you for the plots**
> > >
> > > Thank you! I went back to my original score.
> > >
> > > Good lcuk with other reviewers, I shall keep on reading your discussions.

---

### Official Review · Reviewer_sY4j · 2024-11-04

**Soundness:** 1
**Presentation:** 2
**Contribution:** 1
**Rating:** 3
**Confidence:** 4

**Summary:**

This paper studies RL-guided Branch-and-Bound (B&B) method for Mixed Integer Linear Programs. Specifically, they modeled the procedure of B&B (select which integer variable to do splitting) as a tree MDP which is first proposed by [Scavuzzo et al., 2022]. They proposed a deep q-learning algorithm for the tree MDP, which they named as TreeDQN to guide the variable selection of B&B. Experiments on both synthetic and practical tasks are conducted to show the effectiveness of TreeDQN.

**Strengths:**

1. They did experiments on both synthetic and practical mixed integer linear programming tasks to show the effectiveness of their proposed algorithm.

**Weaknesses:**

The theoretical basis seems not correct to me in this paper which is a big weakness.
1. The definition of value function (2) is not correct. First, the value function (excluding optimal value function) should be policy-dependent. In (2), on the lefthand side, it is policy-independent while on the righthand side, action $a_t$ appears suddenly. Second, the value function shouldn't depend on the node selection strategy since we model the problem as a tree MDP instead of a temporal MDP based on the definition of tree MDP in [Scavuzzo et al., 2022]. Therefore, all the following analysis based on the value function is also under question.
2. The definition of 'contraction in mean' sounds weird to me. What does it mean if an operator is stochastic? And 'contraction in mean' does not guarantee that under this operator, the value function can converge with high probability.
3. The contraction property can be a justification for value iteration methods. However, using it as theoretical backing for q-learning methods is fragile.

**Questions:**

As I have mentioned in the weaknesses part, why does the node selection strategy matter in value function?

---

> ### Author Response · Authors · 2024-11-20
> **Thank you for your time and effort. Here are our answers to the questions and weaknesses.**
>
> > Weakness 1
>
> We apologize for the inconvenience. Certainly, the value function is policy-dependent; the index $\pi$ was skipped for brevity. We will return it back, so the updated version would be:
>
>
> $V_{\pi} (s_t) =$
> $r(s_t, a_t, s_{t+1}) + p^+V_{\pi}(s^{+}_{t+1}) + $
>
> $p^{−}V_{\pi}(s^{-}_{t+1})$
>
>
> However, we cann
> ot agree that the value function shouldn’t depend on the node selection strategy. The value function predicts the expected tree size under the fixed variable selection policy. The node selection policy also influences the resulting size of the tree, thereby influencing the value function. The work [Scavuzzo et al., 2022] also considers a fixed node selection policy. They use a DFS node selection policy in their tmdp+DFS method.
>
> > Weakness 2
>
> When we sample training examples from the replay buffer we want the value function to converge to the optimal value function. We can relax the required contraction property of the Bellman operator to the contraction in mean, which with many training examples, would converge our value function to the optimal value function.
>
>
> > Weakness 3
>
> We agree that the construction property of the Bellman operator does not lead to the contraction of the whole method, which includes a value approximator (graph neural network). In our paper, we never stated that the contraction in mean is the only necessary and sufficient condition to guarantee the convergence of our method. We present it here to motivate our approach and highlight that it has more theoretical foundations than, e.g., FMCTS, one of our benchmarks.
>
>
> > Question 1
>
> The value function estimates the expected return with respect to a policy. In the B&B algorithm, the resulting size of the tree depends on the variable selection policy (which is learned by our RL agent) and the node selection policy (we use the SCIP default node selection policy in our experiments). So, different node selection policies change the resulting size of the tree, thereby influencing the value function.

---

> ### Comment · Reviewer_sY4j · 2024-11-27
>
> Thanks for the clarifications. However, the theoretical parts still look fragile to me.
> 1. I don't think the derivation of the contraction property of the Bellman operator is correct. In line 203, since $p^+, p^-$ is state dependent, when taking the maximum, how could one take them out like they are constant.
> 2. And the authors didn't talk about the optimal Bellman operator either.
> 3. Also, in Scavuzzo et al., 2022, they have assumption 4.2 under which one can form the problem as a tree MDP. In this work, the authors also formulate the problem as tree MDPs while no assumption is claimed.
>
> Besides, there are many mistakes that one shouldn't make.
> 1. $p^+, p^-$ should be state-action dependent. One should express this dependency out explicitly
> 2. In the derivation (line 202-203), $TV-TU$ is a vector, then it suddenly becomes a scalar $p^+V(s^+)+p^-V(s^-)-p^+U(s^+)+p^-U(s^-)$, and when taking the maximum over s, there is even no quantity explicitly dependent on s.
> 3. The authors say for brevity, they skip the policy when defining the value function. It is not appropriate.

---

> > ### Author Response · Authors · 2024-11-28
> > **Thank you for the reply!**
> >
> > You’re correct, we don’t make the same assumptions as in Scavuzzo et.al., 2022. In our work, we use the SCIP default branching heuristic which violates assumption 4.2 of Scavuzzo et.al.  Instead, we make alternative assumptions that $p^{+}$ and $p^{-}$ can be considered independent from the state to some extent as stated in lines 198 - 199. Please note that, while Scavuzzo et.al. formulates theoretical statements and trains the tmdp+DFS agent with the DFS node selection heuristic, they perform an evaluation using the SCIP default node selection heuristic which also violates their theoretical assumptions.
> >
> > The optimal Bellman operator for a TreeMDP is formulated similar to the temporal MDP:
> >
> > $T^*(V(s)) = \max_{a} (r(s, a) + \gamma \left[ p^+V(s^{+}) + p^-V(s^{-}) \right])$
> >
> > We can also show that this operator is a contraction in mean, if $p^{+}$, $p^{-}$ does not depend on state $s$ and action $a$. The proof is constructed similar to the one for the policy Bellman operator for a TreeMDP.
> >
> > Thank you for noticing the typos in the equations. We fixed the issues. Please check out the updated version of our paper.

---

### Official Review · Reviewer_Lqcy · 2024-11-04

**Soundness:** 3
**Presentation:** 3
**Contribution:** 3
**Rating:** 6
**Confidence:** 2

**Summary:**

This paper introduces TreeDQN, an off-policy RL method that enhances the Branch-and-Bound approach for combinatorial optimization by learning efficient branching heuristics. With a proven Bellman contraction for stable training, TreeDQN requires up to 10 times less data and is faster than on-policy methods, achieving superior performance on both synthetic tasks and a challenging ML4CO competition task.

**Strengths:**

1. The paper presents TreeDQN, a novel sample-efficient off-policy RL algorithm designed specifically for combinatorial optimization, which addresses the limitations of high variance and slow training in existing on-policy methods.


2. Theoretical and Empirical Validation: TreeDQN’s theoretical foundation is strong, backed by the contraction property of the Bellman operator for tree MDPs. Empirical results show substantial improvements, including up to 10 times less training data and superior performance on both synthetic and practical tasks, notably the ML4CO competition task.

**Weaknesses:**

1. The experimental section primarily compares TreeDQN with basic methods and lacks extensive benchmarking against a wider range of state-of-the-art approaches in combinatorial optimization, which could better illustrate its relative strengths.


2. The method is tailored to similar MILP tasks, potentially limiting its generalizability to significantly different combinatorial optimization problems, which the paper acknowledges but does not address experimentally.

**Questions:**

1. Could the authors clarify if TreeDQN’s reliance on a contraction property holds under varying conditions or is sensitive to specific task characteristics, such as tree depth or branching factors?

2. What impact does the choice of geometric mean over arithmetic mean have on convergence stability, and are there cases where this choice might be disadvantageous?

3. Could the authors elaborate on how TreeDQN might perform on combinatorial tasks with differing structures, and are there any plans to extend the approach for more varied optimization tasks?

**Details Of Ethics Concerns:**

No concerns

---

> ### Author Response · Authors · 2024-11-20
> **Thank you for your evaluation of our work. Here are our respectful answers to the questions.**
>
> > Weakness 1
>
> The SOTA techniques in combinatorial optimization are usually based on task-independent heuristics that could work for many tasks. Meanwhile, in our work, we consider a set of similar MILP tasks, as stated in the Background section. Hence, we benchmark our reinforcement learning method against similar methods that learn variable selection heuristics for a set of similar tasks, such as tmdp+DFS and FMCTS. We also compare SCIP default, strong branching, and imitation learning methods for a more complete evaluation.
>
> > Weakness 2
>
> The limit of extrapolation to unseen tasks is a common drawback of all learning-based methods. Despite that, with large training data, ML-based methods can show the SOTA results in many practical applications. In our work, we present the results of transferring learned variable selection policies to more complex transfer tasks in Appendix E. It shows that our TreeDQN method can generalize to more complex tasks while trained on simpler tasks.
>
>
> > Question 1
>
> The contraction property holds under a fixed node selection policy. As the background section mentions, the probability of having left and right children $p^+$, $p^-$ depends on the node selection policy. From the theoretical point of view, other task characteristics, such as average tree depth, do not influence the construction property. However, we use function approximators (graph neural networks) to learn a policy that can limit the method's convergence.
>
> > Question 2
>
> Thank you for the important question. We study the choice of the loss function and evaluate TreeDQN with the MSLE and MSE losses in Tab. 2. It shows that our TreeDQN with the MSLE loss function consistently outperforms the competitors. However the learning objectives with MSLE and MSE losses differ. MSLE loss enforces significantly less error on rare large B&B trees (see Fig. 3), hence if the training and evaluation tasks differ and the evaluation tasks produce larger trees than the training ones, the TreeDQN with the MSE loss function could be a better design choice.
>
>
> > Question 3
>
> Thank you for the interesting question. There are different approaches to training a general-purpose heuristic for variable selection:
> 1) If the tasks are not completely different, it would be possible to train a single agent for all tasks or fine-tune the existing agent to a slightly different B&B task.
> 2) Another interesting option is the application of progressive neural networks, allowing the agent to extend the learned policy to the different sets of MILP tasks without forgetting the policy for the initial set of tasks.
>
> Rusu, A. A., Rabinowitz, N. C., Desjardins, G., Soyer, H., Kirkpatrick, J., Kavukcuoglu, K., Pascanu, R., & Hadsell, R. (2016). Progressive Neural Networks (Version 4). arXiv. https://doi.org/10.48550/ARXIV.1606.04671
>
> 3) The third option that may be the most practical is to train a task classifier and learn separate variable selection policies for each class.

---

> > ### Comment · Reviewer_Lqcy · 2024-11-26
> >
> > I appreciate the authors for their rebuttals and explanations. I decide to keep my score.

---

### Note · Authors · 2024-12-10

I have read and agree with the venue's withdrawal policy on behalf of myself and my co-authors.